# Genome-Wide Tiling Array Analysis of HPV-Induced Warts Reveals Aberrant Methylation of Protein-Coding and Non-Coding Regions

**DOI:** 10.3390/genes11010034

**Published:** 2019-12-27

**Authors:** Laith N. AL-Eitan, Mansour A. Alghamdi, Amneh H. Tarkhan, Firas A. Al-Qarqaz

**Affiliations:** 1Department of Applied Biological Sciences, Jordan University of Science and Technology, Irbid 22110, Jordan; amneht92@gmail.com; 2Department of Biotechnology and Genetic Engineering, Jordan University of Science and Technology, Irbid 22110, Jordan; 3Department of Anatomy, College of Medicine, King Khalid University, Abha 61421, Saudi Arabia; m.alghamdi@kku.edu.sa; 4Department of Internal Medicine, Jordan University of Science and Technology, Irbid 22110, Jordan; fqarqaz@just.edu.jo; 5Division of Dermatology, Department of Internal Medicine, King Abdullah University Hospital, Jordan University of Science and Technology, Irbid 22110, Jordan

**Keywords:** HPV, cutaneous warts, methylation, skin

## Abstract

The human papillomaviruses (HPV) are a group of double-stranded DNA viruses that exhibit an exclusive tropism for squamous epithelia. HPV can either be low- or high-risk depending on its ability to cause benign lesions or cancer, respectively. Unsurprisingly, the majority of epigenetic research has focused on the high-risk HPV types, neglecting the low-risk types in the process. Therefore, the main objective of this study is to better understand the epigenetics of wart formation by investigating the differences in methylation between HPV-induced cutaneous warts and normal skin. A number of clear and very significant differences in methylation patterns were found between cutaneous warts and normal skin. Around 55% of the top-ranking 100 differentially methylated genes in warts were protein coding, including the *EXOC4*, *KCNU*, *RTN1*, *LGI1*, *IRF2*, and *NRG1* genes. Additionally, non-coding RNA genes, such as the *AZIN1-AS1*, *LINC02008*, and *MGC27382* genes, constituted 11% of the top-ranking 100 differentially methylated genes. Warts exhibited a unique pattern of methylation that is a possible explanation for their transient nature. Since the genetics of cutaneous wart formation are not completely known, the findings of the present study could contribute to a better understanding of how HPV infection modulates host methylation to give rise to warts in the skin.

## 1. Introduction

Human papillomaviruses (HPV) are double-stranded DNA viruses that exclusively infect the squamous epithelial layers of the mucosa and skin [1]. HPV can be transmitted between individuals via direct contact as well as through fomites, and HPV infection is considered to be the most common sexually transmitted disease on a global scale [2]. Moreover, a number of communicable as well as non-communicable diseases have been associated with HPV infection, including various types of warts and cancers [3]. Hundreds of HPV types have been identified and classified as either low-risk or high-risk depending on their likelihood upon infection to cause malignant or benign symptoms, respectively [4]. Due to their greater carcinogenic potential, the majority of HPV research has centered around high-risk types such as HPV 16 and 18 [5]. In contrast, low-risk HPV infection often manifests in the form of cutaneous warts in otherwise healthy individuals, although it can result in serious pathologies in immunocompromised populations [6].

Cutaneous warts are skin growths that can have different appearances depending on the type and location of HPV infection [7,8]. Common warts, also referred to as *Verruca vulgaris*, are the most prevalent type of wart, constituting around 70% of all non-genital cutaneous warts [9,10]. The wart itself is formed after HPV gains access to the basal layer of the epidermis via a microabrasion in the skin and induces the rapid growth and proliferation of keratinocyte cells [11]. In immunocompetent individuals, HPV-induced warts are self-limiting, as the virus is usually cleared from the body in a period of one to two years [12]. Over the course of the HPV life cycle, the viral genome undergoes dynamic changes in methylation patterns, an observation that has been tentatively attributed to the host’s innate immune response [13]. In fact, HPV methylation is the initial trigger for the transformation of squamous epithelial cells, the latter of which undergo HPV-induced epigenetic re-programming [14]. However, the genetic mechanism of cutaneous wart formation is yet to be clearly elucidated, although their transient nature points towards the involvement of a regulatory epigenetic component such as methylation [15].

Cytosine methylation, which involves the post-replicative modification of CpG dinucleotides, plays a central role in epigenetic regulation by temporarily altering a gene’s functional state [16]. In fact, methylation of CpG sites constitutes the majority of epigenetic modifications, and it is more pronounced in the gene body compared to the transcription promoter regions [17,18]. While promoter methylation results in the silencing of gene expression, gene body methylation is associated with increased expression in dividing cells, and this contradiction, known as the DNA methylation paradox, is not well understood [19,20,21]. However, gene body methylation has been reported to be biologically advantageous as it inhibits the initiation of spurious transcription, the latter of which has undesirable pathological consequences in both lower and higher organisms [22,23].

Maintaining tissue-specific methylation patterns is of great importance to the normal function of many biological processes, as aberrant methylation has been implicated in the tumorigenesis of several types of cancer [24,25,26]. Moreover, aberrant methylation patterns have been implicated in the carcinogenesis of cervical cancer and precancer specimens caused by high-risk HPV infection [13,27,28]. HPV-induced methylation of the host genome has been previously reported for high-risk HPV infection, but, to the best of the author’s knowledge, there is a dearth of information on host methylation in cutaneous warts. In the current study, genomes from wart and normal skin samples were examined by means of a whole-genome tiling array for alterations in methylation profiles.

## 2. Materials and Methods

### 2.1. Study Population and DNA Extraction

Twelve Arab males presenting with HPV-induced warts were recruited at King Abdullah University Hospital (KAUH) in Irbid, Jordan. After obtaining informed consent, paired shave biopsies of the warts (*n* = 12) and nearby normal skin (*n* = 12) were performed by a resident dermatologist. The majority of biopsies were obtained from the hands (*n* = 20), while a few were taken from the forehead (*n* = 2) and foot (*n* = 2). A QIAamp DNA Mini Kit (Qiagen, Hilden, Germany) was used to extract genomic DNA from the 24 tissue samples, and optional RNase A digestion was performed. DNA purity was determined on the BioTek PowerWave XS2 Spectrophotometer (BioTek Instruments, Inc., Winooski, VT, USA), while DNA integrity was ascertained through agarose gel electrophoresis. Samples were then shipped on dry ice to the Australian Genome Research Facility’s (AGRF) Melbourne node for methylation profiling.

### 2.2. Infinium MethylationEPIC BeadChip

At the AGRF, samples were subject to further quality control measures through assessment on the QuantiFluor^®^ dsDNA System (Promega, Gorman, NC, USA) and via 0.8% agarose gel electrophoresis. The 24 samples were then individually made up to approximately 500 ng of DNA in 45 μL, after which they were bisulfite converted by the Zymo EZ DNA Methylation kit (Zymo Research, Irvine, CA, USA). Finally, the samples were individually inputted into the Infinium MethylationEPIC BeadChip microarray (Illumina, San Diego, CA, USA) for interrogation of over 850,000 CpG sites.

### 2.3. Data Processing

A computational R package (RnBeads) was adapted to process and analyze the raw intensity data from the methylation chip in the form of IDAT files [29]. Quality control, preprocessing, batch effects adjustment, and normalization were carried out on all probes and samples according to the RnBeads package pipeline.

### 2.4. Differential Methylation and Statistical Analysis

Differential methylation (DM) analysis was carried out based on tiling region, the latter of which is a genomic region of 5000 bp length. At the genome-tiling level, the mean of the mean β values (mean.mean β) of all the tested probes which span specified genomic regions were computed. The distribution of the number of CpG sites per genome tiling region can be seen in Figure 1A. Additionally, Figure 1B shows the distributions of CpG sites across genome-tiling regions. DM for each genome-tiling region was calculated using three measures: the mean.mean β difference between warts (W) and normal skin (NS); the log2 of the mean quotient in β means across all CpG sites in a genome-tiling region; and the adjusted combined *p*-value of all CpG sites in the genome-tiling region using a limma statistical test [30]. The Benjamini and Hochberg (B–H) 5% false discovery rate (FDR) was used to correct for multiple testing. Additionally, these three measures were used to give each genome-tiling region a combined rank, which was computed as the maximum (=worst) rank among the three ranks. Regions that exhibit more DM will have a smaller combined rank [29]. Genome-tiling regions were sorted from smallest to largest using the combined ranking score, then the best 1000 ranking regions were selected for further analysis.

### 2.5. Locus Overlap Analysis (LOLA) for Enrichment of Genomic Ranges

LOLA was carried out for the top 1000 genome-tiling regions, the latter of which were selected based on combined ranking score [31]. The following LOLA reference databases were utilized in the current analysis: cistrome_cistrome, cistrome_epigenome, codex, encode_segmentation, encode_tfbs, sheffield_dnase, ucsc_features. LOLA uses a Fisher’s exact test to assess the significance of the overlap.

### 2.6. Signaling Pathway Analysis

A signaling network of the genes located in the top 100 most DM genome-tiling regions was created via the Signaling Network Open Resource 2.0 (Signor) [32]. The type of relation was selected to include ‘all’ interactions with a relaxed layout and a score of ‘0.0’.

## 3. Results

### 3.1. Differential Methylation of Genome-Tiling Regions

252,698 genome-tiling regions passed the quality control and pre-processing procedures. Notable differences were seen during the assessment of the methylation level (β) distributions for the genome-tiling regions in wart and normal skin samples (Figure 2). The list of DM genome tiling in warts was limited to the top-ranking 1000 tiling regions using the combined ranking score. Using this scoring method, 772 and 228 tiling regions were found to be hypomethylated and hypermethylated, respectively, in warts compared to normal skin (Figure 3A). Of the 772 hypomethylated tiling regions, the β difference ranged between −0.192 to −0.543, but it ranged between 0.192 and 0.472 in the 228 hypermethylated regions. The log_2_ of the quotient in methylation between warts and normal skin had a maximum value of 2.275 and a minimum value of −2.44 (Figure 3B). The 100 genome-tiling regions with the lowest combined rank scores are presented in Table 1 alongside their gene names.

### 3.2. Clustering of Samples

Samples showed expected clustering based on all methylation values of the top 1000 most variable loci (Figure 4). Samples sharing similar methylation patterns or phenotypes tended to cluster together. In addition, the dataset was subject to a dimension reduction test using multi-dimensional scaling (MDS) in order to inspect for a strong signal in the methylation values of the samples (Figure 5). MDS confirmed that our analysis was dominated by differences between the wart and normal skin samples.

### 3.3. Locus Overlap Analysis (LOLA) for Enrichment of Genomic Ranges

The overall observation of enrichment and overlap across the LOLA reference databases for the hypermethylated and hypomethylated tiling regions are shown in Figure 6 and Figure 7, respectively, while the details of the significantly enriched terms are shown in Figure 8 and Figure 9, respectively. The top-ranking 1000 hypermethylated tiling regions show strong association with the Sheffield_dnase and encode_tfbs databases as indicated by the large odds ratio values. In addition to the strong association with the top tiling regions, encode_tfbs also exhibits a higher statistical significance overlap. From the encode_tfbs database, the c-Fos, STAT3, and c-Myc were among the most enriched terms. Using the Sheffield_dnase database, these tiling regions were predicted to be enriched in several cell and tissue types, including fibroblast cells, epithelial cells, muscle tissue, and skin tissue.

On the other hand, the top-ranking 1000 hypomethylated tiling regions show a strong association with Sheffield_dnase, ucsc_features, and codex indicated by the large odds ratio values, while cistrome_epigenome, encode_tfbe, and encode_segmentation exhibit statistical significance overlap with these tiling regions as indicated by the higher *q*-value. The most enriched terms include the Dnase weak-normal human dermal fibroblast (NHDF) cell line, Dnase-fibrobalsts, Weak Enhancer-human umbilical vein endothelial cells (HUVEC), nuclear receptor subfamily 2 group F member 2 (NR2F2)-Endometrial Stromal Cell, and monomethylation of Histone H3 at lysine 4 (H3K4me1)-lymph node carcinoma of the prostate (LNCaP) cell line.

### 3.4. Pathway Analysis

Signaling network analysis of the genes located in the top 100 DM genome-tiling regions illustrated that two genes, *ATF2* and *HDAC2*, were found to be common regulators of the gene network with a minimum of 12 connectivities each (Figure 10).

## 4. Discussion

Due to their benign nature, low-risk HPV and the cutaneous warts they induce have not been the subject of the same research attention and focus as their high-risk counterparts. In this study, a tiling array was carried out on paired samples of normal skin and warts from 12 Arab males. Methylation profiles were found to significantly differ between the cases and controls, and the top-ranking differentially methylated (DM) genes were mostly protein-coding (55%) and non-coding RNA (ncRNA) (11%) genes.

### 4.1. Aberrant Methylation of Protein-coding Genes

Of the top-ranking 100 DM genes in warts compared to normal skin, 55 were protein-coding genes. The exocyst complex component 4 (*EXOC4*) gene was the most DM protein-coding gene in warts. Not much is known about *EXOC4* except that adjacent polymorphisms were associated with Type 2 diabetes [33]. However, *EXOC4* encodes a component of the exocyst complex, the latter of which is posited to be involved in viral protein transfer between cells [34]. As part of its infection process, HPV relies heavily on membrane-bound transport vesicles to deliver the viral material from the extracellular matrix to the host cell’s nucleus [35,36,37]. The second most DM protein-coding gene is the potassium calcium-activated channel subfamily U member 1 (*KCNU1*) gene, which is a sperm-specific potassium channel that is essential for male fertility [38]. *KCNU1* might be important to HPV biology due to the fact that potassium channels are involved in cell proliferation and apoptosis, among other cellular processes [39]. The reticulon 1 (*RTN1*) gene was the third most DM protein-coding gene in warts, with previous reports showing that RTN1 deficiency and isoforms were associated with senile plaque formation and kidney disease progression, respectively [40,41]. In the context of viral infection, deletion of *RTN1* in yeast cells led to a significant inhibition of viral replication [42].

Interestingly, several DM protein-coding genes, namely the *LGI1*, *IRF2*, and *NRG1* genes, have been implicated in squamous cell carcinoma, which involves the same keratinocyte layer that is exclusively hijacked by HPV infection. The leucine-rich glioma inactivated 1 (*LGI1*) gene is a putative suppressor of metastasis and, when downregulated, was found to stimulate esophageal squamous cell carcinoma metastasis [43]. Similarly, the interferon regulatory factor 2 (*IRF2*) gene, was reported to increase the tumorigenicity of esophageal squamous cell carcinoma when overexpressed [44]. Lastly, the neuregulin 1 (*NRG1*) gene, a cell adhesion molecule, was previously reported to be upregulated in oral squamous cell carcinoma cells [45]. It is important to understand the methylation profiles of benign warts so as to improve the understanding of the effects of low-risk HPV infection on host methylation, the latter of which has been extensively explored in the context of high-risk HPV infection.

Furthermore, various DM protein-coding genes in warts have been previously associated with cancers other than squamous cell carcinoma. Encoding an argonaute family protein, the piwi-like protein 4 (*PIWIL4*) gene was reported to be highly expressed in breast cancer cells, and its knockdown was found to lessen leukemic growth [46,47]. Moreover, increased expression of the protein tyrosine phosphatase (*PTPRA*) gene and promoter methylation of the calcium-binding protein 39-like (*CAB39L*) gene were associated with gastric cancer [48,49]. In addition, the abundant expression of the solute carrier family 22 member 16 (*SLC22A16*) gene, a carnitine transporter, has been reported in ovarian carcinoma cell lines, and its upregulation helped induce melanoma cell death when combined with chemotherapy [50,51]. Likewise, the dimethylarginine dimethylaminohydrolase 1 (*DDAH1*) gene, which plays an integral role in methylarginine removal, was found to be frequently upregulated in prostate cancer [52], downregulated in gastric cancer [53], and inhibited in attenuated triple negative breast cancer cells [54]. In contrast, high expression of the Kelch-like family member 7 (*KLHL7*) gene, which encodes for a mediator of ubiquitination, is associated with aggressive breast cancer progression [55]. The fact that some protein-coding genes are DM in both warts and various HPV-associated cancers could potentially suggest that the extent of methylation contributes to whether the phenotype is malignant or benign.

### 4.2. Aberrant Methylation of Non-Coding Genes

Non-coding RNAs (ncRNAs) are non-protein coding RNA molecules that generally do not possess a known biological function, although a small minority have been identified as having important functional roles [56]. Dysregulated ncRNA expression patterns have also been implicated in HPV-associated cancers caused by high-risk HPV infection [57]. 11 of the top-ranking 100 DM genes in warts were non-coding RNAs (ncRNAs), including the most DM gene in warts, *AZIN1-AS1*. Very little is known about the AZIN1 antisense RNA 1 (*AZIN1-AS1*) gene both in terms of its function and disease associations. Likewise, the second and third most DM ncRNAs in warts were the long intergenic non-protein coding RNA 2008 (*LINC02008*) and the uncharacterized MGC27382 (*MGC27382*) genes. The *MGC27382* gene was previously reported to be a part of an endogenous RNA network that could serve as a prognostic biomarker for lung squamous cell carcinoma [58]. Moreover, *MGC27382* was found to be significantly upregulated in colorectal cancer but downregulated in lung adenocarcinoma [59,60]. In addition, the long intergenic non-protein coding RNA 2241 (LINC02241) gene was the fourth most DM ncRNA in warts and was previously associated with hepatocellular and colorectal carcinomas [61,62]. The fifth most DM ncRNA was the FER1L6 antisense RNA 1 (*FER1L6-AS1*) gene, which was found to be dysregulated in esophageal squamous cell carcinoma [63].

### 4.3. Genomic Hypermethylation

Hypermethylation of DNA has been associated with transcriptional silencing of the affected genes. From among ENCODE’s transcription factor binding sites, the *c-Fos*, *STAT3*, and *c-Myc* genes were the most hypermethylated in warts (Figure 7). The proto-oncogene *c-Fos* is involved in cell differentiation and proliferation, and its expression is required for skin tumors to become malignant [64]. Moreover, the induction of *c-Fos* expression promotes inflammation of the skin that, in turn, mediates the development of preneoplastic lesions [65]. On the other hand, *c-Fos* expression was found to decrease keratinocyte growth by increasing sensitivity to apoptosis, but this state was reversed upon the addition of c-Jun [66]. In high-risk HPV infection, skin tumorigenesis was found to be critically dependent on E2-induced *c-Fos* expression [67].

Similarly, the signal transducer and activator of transcription 3 (*STAT3*) gene is a transcription factor that is essential for cell apoptosis and growth [68]. In the skin, *STAT3* has a dual role in maintaining homeostasis and wound-healing as well as promoting carcinogenesis and psoriasis when aberrantly expressed [69,70]. During periods of bacterial and viral infection, *STAT3* creates an inflammatory microenvironment that induces carcinogenesis [71]. *STAT3* activation is a common mechanism of herpesvirus pathogenesis, especially in the context of the varicella-zoster virus [72]. Additionally, autocrine *STAT3* activation is an integral part of HPV-mediated cervical cancer, and loss of *STAT3* expression is detrimental to high-risk HPV infection of keratinocytes [73,74].

Lastly, the *c-Myc* gene is an oncogenic transcription factor that regulates the expression of 15% of the human genome and is dysregulated in the majority of human cancers [75]. In the mammalian epidermis, *c-Myc* overexpression helps stimulate keratinocyte proliferation, while *c-Myc* knockdown results in the latter’s inhibition [76]. Alongside the SIN3 transcription regulator family member A (*SIN3A*) gene, *c-Myc* helps maintain tissue homeostasis in the skin, but epidermal cells deficient in c-Myc were found to be resistant to Ras-mediated tumorigenesis [77,78]. Furthermore, *c-Myc* gene amplification was highly associated with infection by oncogenic high-risk HPV types in cervical carcinomas [79].

### 4.4. Genomic Hypomethylation

In contrast to hypermethylation, hypomethylation of genomic DNA often leads to the activation of the affected genes. In warts, the *AR*, *NR2F2*, and *AFF1* genes were among the most significantly hypomethylated in warts compared to normal skin (Figure 9). The androgen receptor (*AR*) gene, which is a nuclear receptor activated by androgenic hormones, normally functions as a DNA-binding transcription factor that is important in the development of male sexual characteristics [80]. Moreover, *AR* upregulation has been reported to suppress wound healing and increase inflammation in a murine model, and its dysregulation is involved in a number of different skin pathologies [81]. Loss of *AR* expression was reported to be a common event in intraepithelial neoplasia and invasive squamous cell carcinoma associated with high-risk HPV infection of the cervix [82].

Likewise, the nuclear receptor subfamily 2 (*NR2F2*) and the AF4/FMR2 family member 1 (*AFF1*) genes encode for nuclear transcription factors that play a critical role in vascular development and osteogenic differentiation, respectively [83,84]. *NR2F2* inhibition by miR-302 contributes to somatic cell pluripotency, and its expression plays an important role in the chondrogenesis of mesenchymal stem cells [85,86]. In addition, the *AFF1* gene was found to be downregulated in melanoma tissue and dysregulated in acute lymphoblastic leukemia [87,88]. The involvement of the *NR2F2* and *AFF1* genes in HPV infection is still not clear.

### 4.5. Genes Involved in the Signaling Network Pathway

Two genes, *ATF2* and *HDAC2*, were found to be common regulators of the top-ranking 100 most DM genes in warts (Figure 10). Also known as cyclic AMP response element binding protein 2 (*CREB2*), the activating transcription factor 2 (*ATF2*) gene was found to be a common regulator of the DM gene network in HPV-induced warts. *ATF2* encodes a leucine zipper transcription factor that can also act as a histone acetyltransferase, and it has been implicated in various malignant skin diseases [89]. In fact, *ATF2* overexpression was necessary for tumor growth and progression in murine skin and for melanoma metastasis in human skin [90,91]. On a similar note, the histone deacetylase 2 (*HDAC2*) gene, which encodes for an enzyme that deacetylates lysine residues situated within core histone N-terminal regions, plays an essential role in epidermal development [92]. *HDAC2* inhibition was found to stabilize tumor suppression and induce apoptosis in human keratinocyte cells infected with high-risk HPV [93].

### 4.6. Anatomical Location of Warts

In the present study, most of the warts were obtained from the hands (*n* = 20), with a minority taken from the feet (*n* = 2) and forehead (*n* = 2). Anatomic site is an important factor to consider in such studies, as it can influence the expression and methylation of genes. The hands and forehead, for example, are exposed to environmental factors that other parts of the body are not, leading to differences in gene expression between exposed and non-exposed skin [94]. Interestingly, in a cancer context, one study reported a novel epigenetic signature of HPV infection that was independent of the anatomic location in HPV-associated head and neck squamous cell carcinomas [95].

## 5. Conclusions

Wart formation was found to involve a clear methylation pattern that sets it apart from normal skin. It was demonstrated that most differentially methylated genes were protein coding and included non-coding RNA genes as well. Surprisingly, many of the DM genes found in benign warts were previously reported to be involved in high-risk HPV infection and HPV-associated malignancies. The main limitation of the present study was that the participants were all male, but only males were included in order to minimize any genetic variation that might occur due to differences in sex.

## Figures and Tables

**Figure 1 genes-11-00034-f001:**
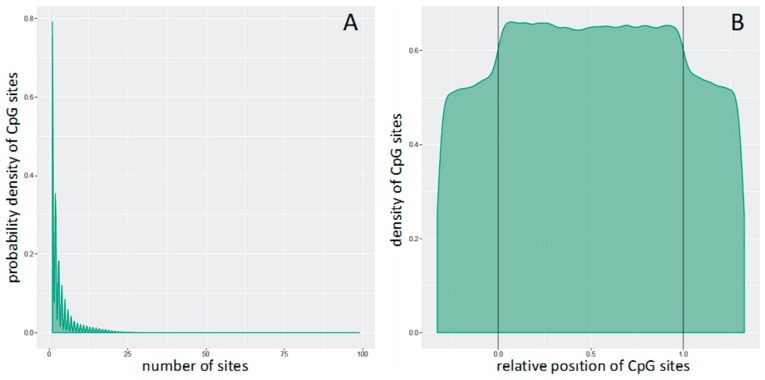
Distributions of CpG sites (**A**) per genomic tiling region and (**B**) across genomic tiling region. In Panel B, the relative coordinates of 0 and 1 correlate to the start and end coordinates of the genomic tiling region. Those coordinates that are smaller than 0 and larger than 1 indicate flanking regions normalized by region length.

**Figure 2 genes-11-00034-f002:**
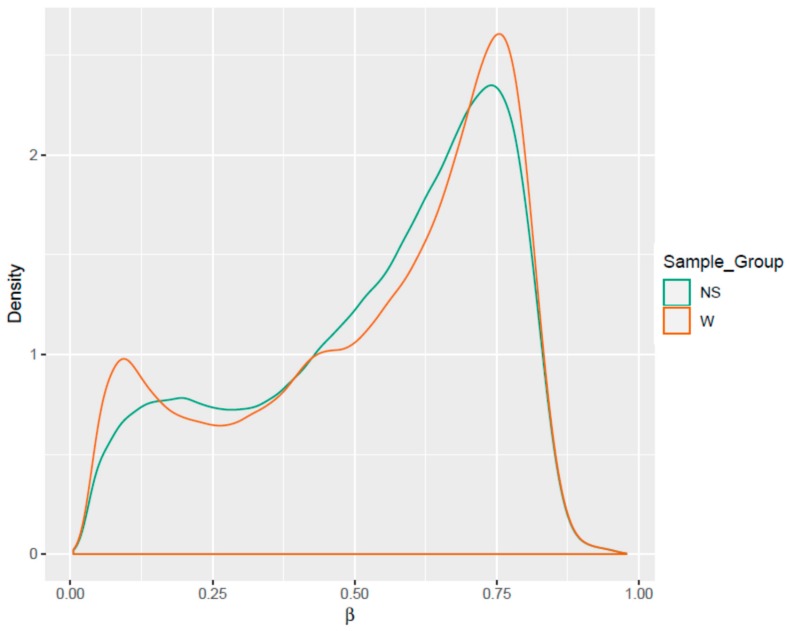
Comparisons of the density distributions of methylation levels (β) in warts (W) and normal skin (NS).

**Figure 3 genes-11-00034-f003:**
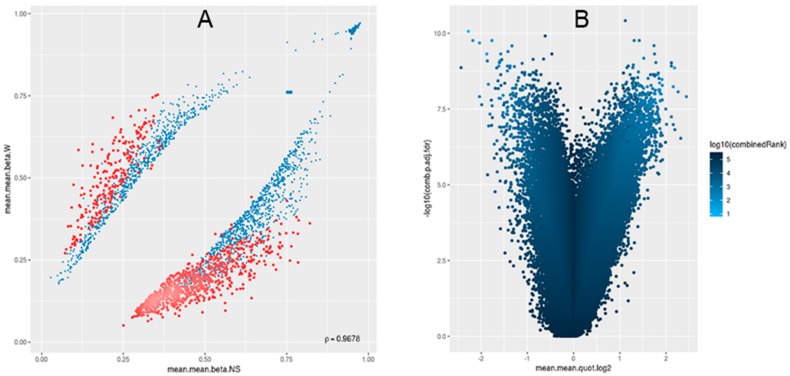
For the 1000 most differentially methylated tiling regions, scatterplot (**A**) and (**B**) volcano plot analyses were performed. In panel A, the mean of mean methylation levels (β) for warts (W) is on the y-axis, while the mean of mean methylation levels (β) for normal skin (NS) is on the x-axis. β values range from unmethylated (0) to methylated (1). In panel B, the volcano plot shows the differential methylation of genomic tiling regions as quantified by log_2_ of the mean quotient in means across all sites in a region on the x-axis and the adjusted combined p-value on the y-axis between warts (W) and normal skin (NS). The color scale corresponds with the combined rank of each genomic tiling region.

**Figure 4 genes-11-00034-f004:**
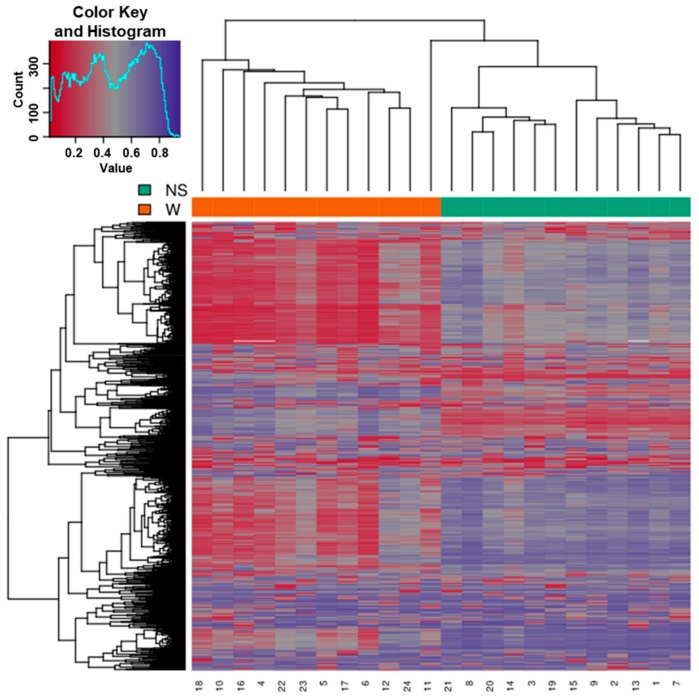
Heatmap showing the hierarchical clustering of samples displaying only the 1000 most variable loci with the highest variance across all samples. Clustering used complete linkage and Manhattan distance. Patient identification number is shown on the bottom *x*-axis. The normal skin (NS) and wart (W) samples are shown on the top *x*-axis. Values of 0 (red color) and 1 (purple color) indicate decreased and increased methylation, respectively.

**Figure 5 genes-11-00034-f005:**
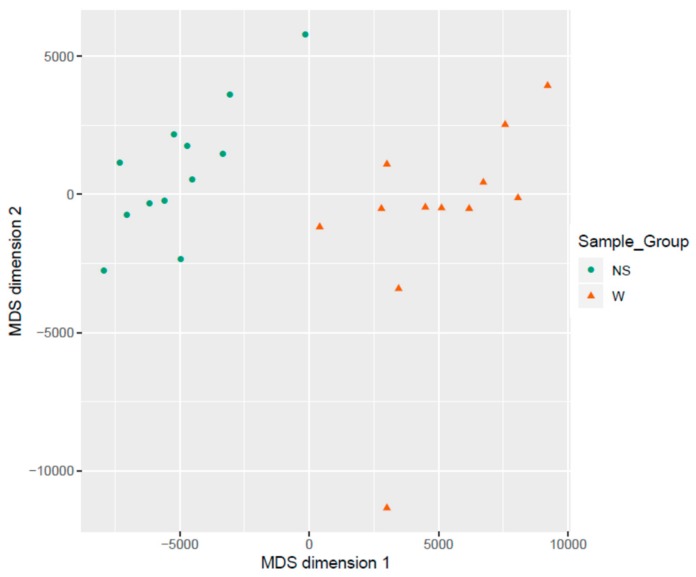
Scatter plot showing samples after performing Kruskal’s non-metric multidimensional scaling based on the matrix of average methylation levels and Manhattan distance.

**Figure 6 genes-11-00034-f006:**
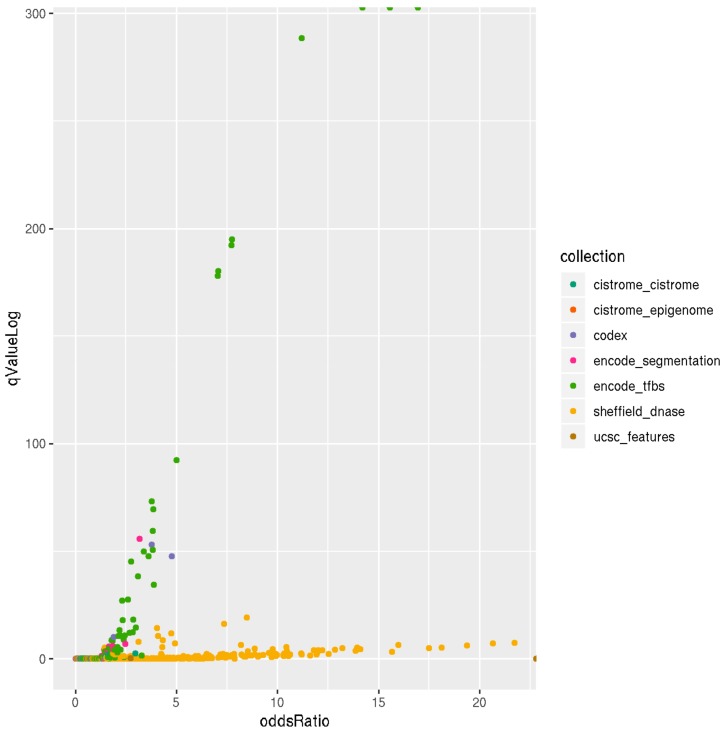
Scatterplot of LOLA enrichment analysis showing the effect size (log-odds ratio) versus the significant *q*-value (−log10 (*q*-value)) of the 1000 most hypermethylated tiling regions. These regions show strong association and significant overlap with Sheffield_dnase and encode_tfbs as indicated by the large odds ratio and higher *q*-value. LOLA reference databases collections are shown on the right side of the plot with color coding.

**Figure 7 genes-11-00034-f007:**
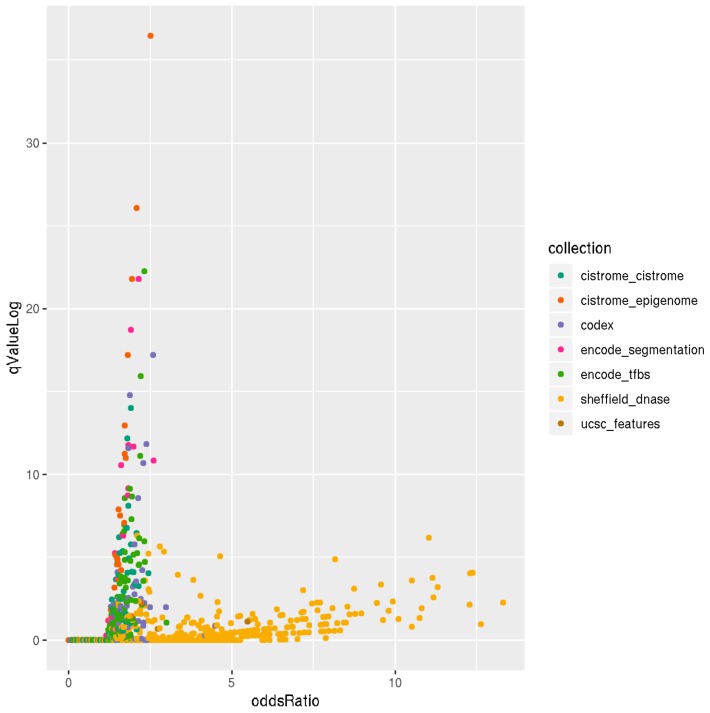
Scatter plot of LOLA enrichment analysis showing the effect size (log-odds ratio) versus the significant *q*-value (−log10 (*q*-value)) of the 1000 most hypomethylated tiling regions. These regions show strong association and significant overlap with Sheffield_dnase, ucsc_features, codex, cistrome_epigenome, encode_tfbe, and encode_segmentation as indicated by the large odds ration values and the higher q−value. LOLA reference databases collections are shown on the right side of the plot with color coding.

**Figure 8 genes-11-00034-f008:**
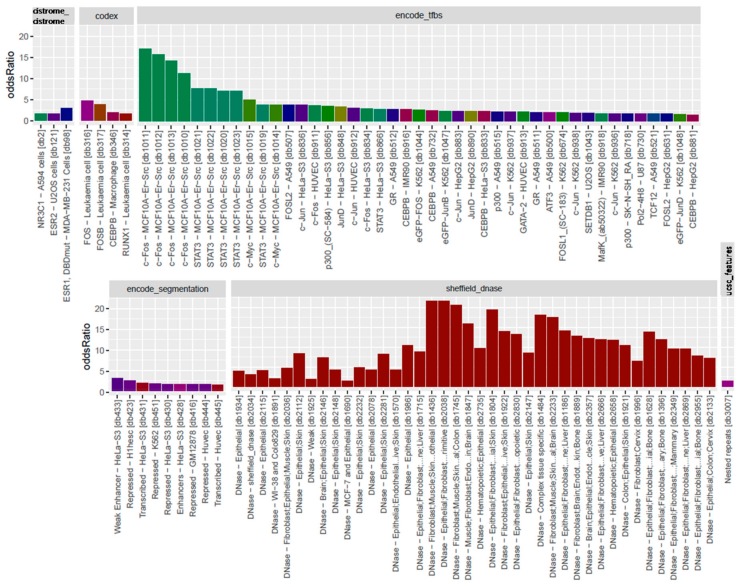
Bar plots of LOLA enrichment analysis showing the log-odds ratios of the 1000 most hypermethylated tiling regions. Terms that exhibit statistical significance (*p*-value < 0.01) are shown. For encode_tfbs, c-Fos, STAT3, and c-Myc are among the top enriched terms showing large odds ratio values. Fibroblast cells, epithelial cells, muscle tissue, and skin tissue were among the most enriched cell and tissue types. Coloring of the bars reflects the putative targets of the terms.

**Figure 9 genes-11-00034-f009:**
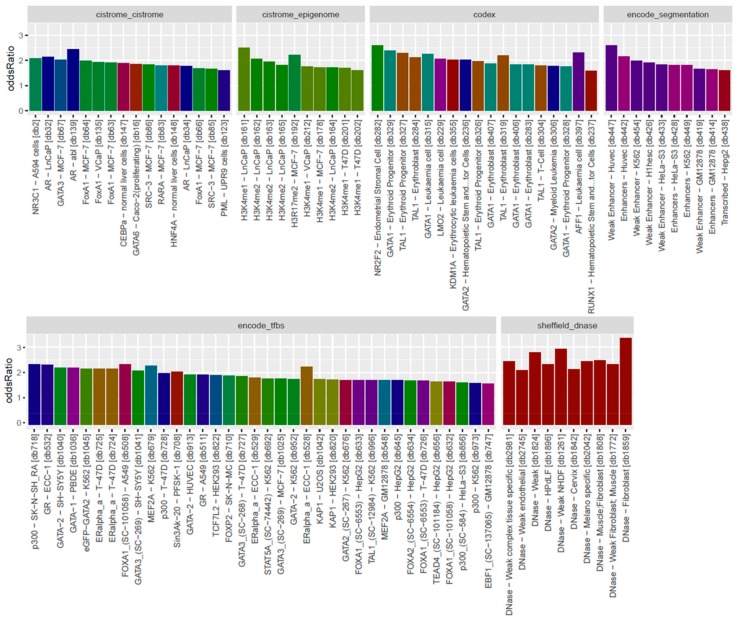
Bar plots of LOLA enrichment analysis showing log-odds ratios of the 1000 most hypomethylated tiling regions. Terms that exhibit statistical significance (*p*-value < 0.01) are shown. The most enriched terms include Dnase weak-NHDF, Dnase-fibrobalsts, Weak Enhancer-HUVEC, NR2F2-Endometrial Stromal Cell, AFF1-leukaemia, H3K4me1-LNCaP, and androgen receptor (AR)-abl. Coloring of the bars reflects the putative targets of the terms.

**Figure 10 genes-11-00034-f010:**
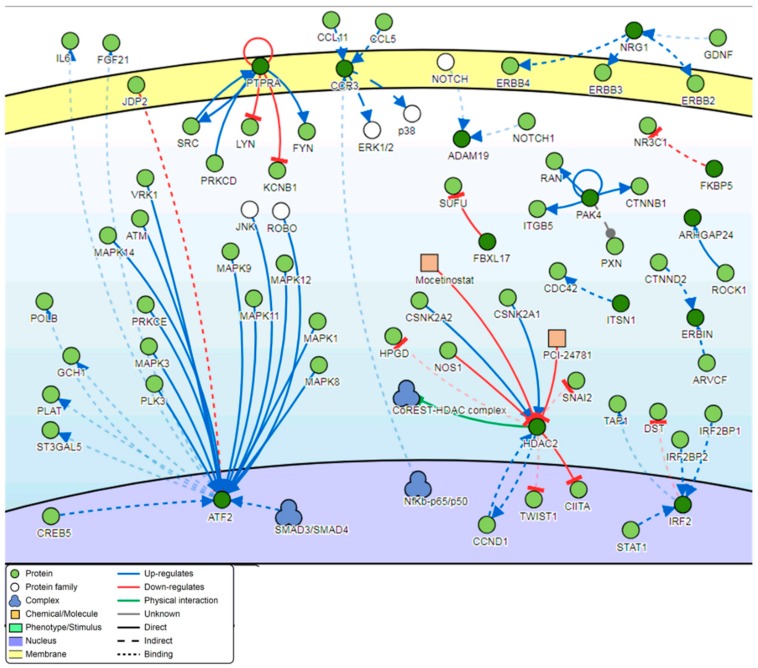
Pathway signaling network generated from the genes located within the top 100 DM tiling regions.

**Table 1 genes-11-00034-t001:** Top-ranking 100 differentially methylated genes in warts (W) compared to normal skin (NS).

Gene	Category	Chromosome	Start	End	Mean.mean β Value (NS)	Mean.mean β Value (W)	Mean.mean β Value Difference (W-NS)	Mean.mean. quot.log2	Comb.p.val	Comb.p.adj.fdr	Combined Rank
		10	104555001	104560000	0.113	0.585	0.472	2.275	6.818 × 10^−16^	8.614 × 10^−11^	8
*AZIN1-AS1*	RNA gene (ncRNA)	8	102975001	102980000	0.157	0.603	0.446	1.878	2.736 × 10^−14^	4.816 × 10^−10^	32
*EXOC4*	Protein coding	7	133390001	133395000	0.642	0.134	−0.509	−2.184	2.139 × 10^−13^	1.386 × 10^−9^	39
		10	4280001	4285000	0.125	0.538	0.412	2.031	6.565 × 10^−15^	2.074 × 10^−10^	48
*KCNU1*	Protein coding	8	36840001	36845000	0.583	0.147	−0.436	−1.914	9.147 × 10^−13^	3.350 × 10^−9^	69
*SVIL2P*	Pseudogene	10	30675001	30680000	0.617	0.163	−0.454	−1.859	1.342 × 10^−12^	4.403 × 10^−9^	77
*RTN1*	Protein coding	14	59720001	59725000	0.151	0.538	0.387	1.768	2.738 × 10^−15^	1.730 × 10^−10^	95
		4	62330001	62335000	0.561	0.118	−0.443	−2.153	2.391 × 10^−12^	5.796 × 10^−9^	104
*TBC1D22A*	Protein coding	22	47045001	47050000	0.098	0.479	0.381	2.180	3.913 × 10^−15^	1.732 x 10^−10^	110
*PIWIL4, AP000943.3*	Protein coding, RNA gene	11	94610001	94615000	0.609	0.162	−0.446	−1.843	3.742 × 10^−12^	7.253 × 10^−9^	130
*LINC02008*	RNA gene (ncRNA)	3	82205001	82210000	0.524	0.137	−0.387	−1.863	3.998 × 10^−12^	7.488 10^−9^	134
*LGI1*	Protein coding	10	93770001	93775000	0.514	0.135	−0.379	−1.856	4.742 × 10^−12^	8.207 × 10^−9^	146
*VPS16, PTPRA*	Protein coding	20	2865001	2870000	0.499	0.134	−0.364	−1.819	5.040 × 10^−13^	2.318 × 10^−9^	151
		5	115915001	115920000	0.513	0.132	−0.381	−1.879	6.570 × 10^−12^	9.941 × 10^−9^	167
*DPCD*	Protein coding	10	101590001	101595000	0.579	0.170	−0.409	−1.758	6.785 × 10^−12^	1.005 × 10^−8^	170
		12	105855001	105860000	0.497	0.126	−0.370	−1.893	6.837 × 10^−12^	1.005 × 10^−8^	171
*MGC27382*	RNA gene (ncRNA)	1	78295001	78300000	0.185	0.586	0.402	1.615	1.534 × 10^−13^	1.211 × 10^−9^	177
*IRF2*	Protein coding	4	184410001	184415000	0.483	0.125	−0.357	−1.864	4.053 × 10^−12^	7.531 × 10^−9^	182
*SLC22A16*	Protein coding	6	110460001	110465000	0.475	0.118	−0.357	−1.916	2.021 × 10^−12^	5.283 × 10^−9^	186
		2	152330001	152335000	0.171	0.536	0.365	1.593	7.112 × 10^−14^	8.169 × 10^−10^	195
*CAB39L*	Protein coding	13	49365001	49370000	0.563	0.095	−0.467	−2.441	9.568 × 10^−12^	1.229 × 10^−8^	195
		2	220830001	220835000	0.650	0.197	−0.454	−1.677	1.176 × 10^−11^	1.363 × 10^−8^	218
		3	24080001	24085000	0.218	0.684	0.466	1.607	1.218 × 10^−11^	1.406 × 10^−8^	219
*NRG1*	Protein coding	8	32280001	32285000	0.171	0.530	0.360	1.580	1.266 × 10^−11^	1.434 × 10^−8^	223
*FAAHP1*	Pseudogene	1	46465001	46470000	0.507	0.159	−0.348	−1.613	5.414 × 10^−12^	9.001 × 10^−9^	228
*L3HYPDH, AL121694.1, JKAMP*	Protein coding, RNA gene, Protein coding	14	59480001	59485000	0.494	0.139	−0.355	−1.756	1.410 × 10^−11^	1.543 × 10^−8^	231
*BPIFA4P*	Pseudogene	20	33205001	33210000	0.785	0.243	−0.543	−1.655	1.472 × 10^−11^	1.569 × 10^−8^	237
*CUZD1*	Protein coding	10	122835001	122840000	0.124	0.469	0.346	1.843	1.675 × 10^−12^	4.809 × 10^−9^	242
*LINC02241*	RNA gene (ncRNA)	5	20675001	20680000	0.506	0.148	−0.358	−1.707	1.547 × 10^−11^	1.608 × 10^−8^	243
		6	14050001	14055000	0.530	0.136	−0.394	−1.889	1.587 × 10^−11^	1.630 × 10^−8^	246
*FER1L6, FER1L6-AS1*	Protein coding, RNA gene (ncRNA)	8	123990001	123995000	0.503	0.160	−0.344	−1.596	2.420 × 10^−12^	5.796 × 10^−9^	251
*DDAH1, AC092807.3*	Protein coding, RNA gene	1	85555001	85560000	0.549	0.143	−0.407	−1.873	1.752 × 10^−11^	1.710 × 10^−8^	258
		15	86070001	86075000	0.172	0.515	0.342	1.526	8.259 × 10^−12^	1.116 × 10^−8^	266
		1	232655001	232660000	0.421	0.080	−0.341	−2.263	5.031 × 10^−12^	8.532 × 10^−9^	269
*KLHL7*	Protein coding	7	23135001	23140000	0.525	0.150	−0.375	−1.738	1.938 × 10^−11^	1.814 × 10^−8^	270
*TANGO6*	Protein coding	16	69070001	69075000	0.614	0.208	−0.406	−1.519	4.213 × 10^−14^	6.654 × 10^−10^	273
*LINC01090*	RNA gene (ncRNA)	2	188220001	188225000	0.569	0.193	−0.376	−1.510	3.271 × 10^−12^	6.621 × 10^−9^	285
*CASC2*	RNA gene (ncRNA)	10	118190001	118195000	0.485	0.142	−0.343	−1.700	2.307 × 10^−11^	1.971 × 10^−8^	295
		20	5040001	5045000	0.685	0.236	−0.450	−1.500	2.152 × 10^−11^	1.915 × 10^−8^	298
*ARHGAP24*	Protein coding	4	85610001	85615000	0.500	0.153	−0.346	−1.640	2.522 × 10^−11^	2.096 × 10^−8^	304
*CCR3*	Protein coding	3	46235001	46240000	0.542	0.188	−0.354	−1.480	8.227 × 10^−12^	1.116 × 10^−8^	322
*THOC2*	Protein coding	X	123630001	123635000	0.652	0.208	−0.444	−1.603	3.058 × 10^−11^	2.321 × 10^−8^	333
*FBXL17*	Protein coding	5	108135001	108140000	0.717	0.221	−0.497	−1.658	3.152 × 10^−11^	2.351 × 10^−8^	338
*VPS13D, SNORA59A*	Protein coding, RNA gene (snoRNA)	1	12505001	12510000	0.190	0.517	0.328	1.576	2.859 × 10^−14^	4.816 × 10^−10^	346
		5	141905001	141910000	0.491	0.166	−0.326	−1.513	3.232 × 10^−12^	6.621 × 10^−9^	357
		1	173115001	173120000	0.491	0.164	−0.327	−1.523	3.557 × 10^−11^	2.495 × 10^−8^	358
		5	57295001	57300000	0.576	0.174	−0.401	−1.669	3.617 × 10^−11^	2.525 × 10^−8^	362
*ERBIN*	Protein coding	5	66010001	66015000	0.495	0.161	−0.334	−1.563	3.822 × 10^−11^	2.592 × 10^−8^	371
*AC105758.1*	Pseudogene	4	126105001	126110000	0.468	0.145	−0.324	−1.628	1.828 × 10^−11^	1.743 × 10^−8^	374
		17	25850001	25855000	0.206	0.573	0.367	1.431	1.336 × 10^−13^	1.206 × 10^−9^	410
*FAM76B*	Protein coding	11	95775001	95780000	0.226	0.623	0.397	1.429	2.820 × 10^−13^	1.738 × 10^−9^	415
		1	209490001	209495000	0.635	0.201	−0.434	−1.614	5.365 × 10^−11^	3.172 × 10^−8^	426
*EFCAB13, AC040934.1*	Protein coding, RNA gene	17	47415001	47420000	0.092	0.410	0.317	2.034	9.620 × 10^−12^	1.229 × 10^−8^	430
*PHC2*	Protein coding	1	33405001	33410000	0.515	0.186	−0.329	−1.422	2.189 10^−12^	5.477 × 10^−9^	435
*AGO4*	Protein coding	1	35835001	35840000	0.203	0.559	0.356	1.418	4.086 × 10^−11^	2.675 × 10^−8^	443
*AC079160.1*	RNA gene	4	84235001	84240000	0.503	0.163	−0.341	−1.572	5.979 × 10^−11^	3.410 × 10^−8^	443
*PAK4*	Protein coding	19	39160001	39165000	0.180	0.498	0.318	1.417	1.745 × 10^−14^	4.009 × 10^−10^	448
*AP003100.2*	RNA gene	11	112700001	112705000	0.510	0.185	−0.325	−1.413	2.551 × 10^−11^	2.114 × 10^−8^	456
*CLEC4C*	Protein coding	12	7750001	7755000	0.433	0.119	−0.314	−1.777	2.365 × 10^−11^	1.998 × 10^−8^	456
		12	92635001	92640000	0.222	0.614	0.392	1.429	6.703 × 10^−11^	3.642 × 10^−8^	465
		6	164210001	164215000	0.475	0.137	−0.337	−1.717	6.867 × 10^−11^	3.716 × 10^−8^	467
*METTL15*	Protein coding	11	28160001	28165000	0.724	0.267	−0.456	−1.404	3.133 × 10^−13^	1.885 × 10^−9^	477
*RTKN2*	Protein coding	10	62205001	62210000	0.158	0.474	0.315	1.522	7.807 × 10^−11^	4.119 × 10^−8^	479
*LINC00824*	RNA gene (ncRNA)	8	128535001	128540000	0.439	0.127	−0.312	−1.709	3.343 × 10^−12^	6.660 × 10^−9^	482
*MSANTD3-TMEFF1, TMEFF1*	Protein coding	9	100550001	100555000	0.564	0.209	−0.355	−1.392	1.378 × 10^−12^	4.408 × 10^−9^	500
*MBNL1*	Protein coding	3	152310001	152315000	0.121	0.447	0.326	1.804	9.367 × 10^−11^	4.669 × 10^−8^	507
*AC092106.2*	Pseudogene	2	106205001	106210000	0.486	0.170	−0.316	−1.461	9.531 × 10^−11^	4.677 × 10^−8^	515
		13	106570001	106575000	0.424	0.113	−0.311	−1.822	9.607 × 10^−11^	4.705 × 10^−8^	516
*FKBP5*	Protein coding	6	35690001	35695000	0.219	0.605	0.386	1.424	1.015 × 10^−10^	4.858 × 10^−8^	528
*ARFGAP3*	Protein coding	22	42805001	42810000	0.243	0.647	0.405	1.379	2.067 × 10^−13^	1.386 × 10^−9^	529
*DDAH1, AL078459.1*	Protein coding, RNA gene	1	85370001	85375000	0.456	0.150	−0.306	−1.540	1.696 × 10^−11^	1.689 × 10^−8^	540
*HDAC2*	Protein coding	6	113930001	113935000	0.241	0.640	0.398	1.371	1.476 × 10^−13^	1.211 × 10^−9^	545
		10	4925001	4930000	0.427	0.123	−0.304	−1.719	3.638 × 10^−11^	2.533 × 10^−8^	555
*AC011287.1*	RNA gene	7	13235001	13240000	0.429	0.125	−0.304	−1.699	1.148 × 10^−10^	5.173 × 10^−8^	562
		8	92330001	92335000	0.476	0.164	−0.312	−1.479	1.154 × 10^−10^	5.173 × 10^−8^	564
*CD96*	Protein coding	3	111620001	111625000	0.468	0.153	−0.316	−1.557	1.178 × 10^−10^	5.233 × 10^−8^	569
		8	8945001	8950000	0.456	0.143	−0.313	−1.607	1.183 × 10^−10^	5.233 × 10^−8^	571
*MRPL33, BABAM2*	Protein coding	2	27980001	27985000	0.467	0.165	−0.302	−1.449	1.378 × 10^−11^	1.514 × 10^−8^	573
*TEX15*	Protein coding	8	30855001	30860000	0.561	0.212	−0.349	−1.361	1.055 x 10^−10^	4.956 x 10^−8^	574
		14	51125001	51130000	0.116	0.417	0.301	1.763	3.558 × 10^−11^	2.495 × 10^−8^	581
*AC008676.3, ADAM19*	Protein coding	5	157425001	157430000	0.598	0.160	−0.437	−1.835	1.322 × 10^−10^	5.680 × 10^−8^	588
		4	184025001	184030000	0.433	0.133	−0.300	−1.634	8.532 × 10^−11^	4.382 × 10^−8^	590
*COG2*	Protein coding	1	230645001	230650000	0.409	0.087	−0.323	−2.117	1.358 × 10^−10^	5.756 × 10^−8^	596
*ATF2*	Protein coding	2	175125001	175130000	0.414	0.115	−0.300	−1.767	4.230 × 10^−12^	7.608 × 10^−9^	599
		7	134975001	134980000	0.406	0.107	−0.298	−1.825	5.356 × 10^−11^	3.172 × 10^−8^	610
*LINC01320*	RNA gene (ncRNA)	2	34325001	34330000	0.525	0.156	−0.369	−1.691	1.510 × 10^−10^	6.206 × 10^−8^	615
*ADAMTS6*		5	65405001	65410000	0.443	0.134	−0.309	−1.654	1.531 × 10^−10^	6.251 × 10^−8^	619
*AC013356.1*	Pseudogene	15	40480001	40485000	0.432	0.129	−0.303	−1.667	1.554 × 10^−10^	6.314 × 10^−8^	621
*AL050403.2*	RNA gene	20	10735001	10740000	0.554	0.171	−0.383	−1.640	1.555 × 10^−10^	6.314 × 10^−8^	622
*MANSC1*	Protein coding	12	12345001	12350000	0.482	0.172	−0.310	−1.434	1.571 × 10^−10^	6.354 × 10^−8^	625
*UAP1*	Protein coding	1	162575001	162580000	0.410	0.114	−0.296	−1.762	6.338 × 10^−12^	9.887 × 10^−9^	628
*AP000311.1, ITSN1*	Protein coding	21	33710001	33715000	0.408	0.112	−0.296	−1.773	9.062 × 10^−11^	4.580 × 10^−8^	630
*NRG1*	Protein coding	8	32550001	32555000	0.421	0.125	−0.296	−1.675	1.996 × 10^−12^	5.283 × 10^−9^	632
*LINC01470*	RNA gene (ncRNA)	5	152940001	152945000	0.483	0.160	−0.323	−1.538	1.668 × 10^−10^	6.602 × 10^−8^	638
*FOXN3*	Protein coding	14	89165001	89170000	0.566	0.208	−0.359	−1.404	1.672 × 10^−10^	6.602 × 10^−8^	640
*GPM6B*	Protein coding	X	13840001	13845000	0.528	0.152	−0.376	−1.730	1.729 × 10^−10^	6.795 × 10^−8^	643
		7	158605001	158610000	0.200	0.519	0.319	1.333	3.342 × 10^−11^	2.427 × 10^−8^	651
		12	59475001	59480000	0.273	0.687	0.414	1.332	1.031 × 10^−14^	2.605 × 10^−10^	653
*FOXN3*	Protein coding	14	89345001	89350000	0.479	0.131	−0.348	−1.796	1.893 × 10^−10^	7.201 × 10^−8^	664
*LINC01098*	RNA gene (ncRNA)	4	177930001	177935000	0.609	0.212	−0.397	−1.478	1.914 × 10^−10^	7.261 × 10^−8^	666

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
