# Peer review of "Genome-Wide Tiling Array Analysis of HPV-Induced Warts Reveals Aberrant Methylation of Protein-Coding and Non-Coding Regions"

_genes, 2019, doi:10.3390/genes11010034_

Round 1
Reviewer 1 Report
This manuscript describes a genome wide analysis in tissue samples from low-risk HPV-induced warts in comparison with surrounding normal tissue. The authors report genes that are either hypermethylated or hypomethylated in warts tissue when compared to normal tissue.
This study provides valuable insight on the epigenetics of cutaneous warts and describes significant methylation pattern differences that could shed light to the understanding of host modulation by HPV infection. The authors handled highly complexed genomic datasets analyzed by R software packages. The findings sound interesting and worth exploring its biological mechanisms, but some explanations are lacking and some figures could become supplemental due to its redundancy. Also, the manuscript would benefit from a more detailed discussion focusing on exploring explanations that would clarify the findings. This reviewer feels it would improve greatly this manuscript if the discussion section would be restructured. The study is innovative and reports good findings that would get higher visibility with explorative associations.
Lines 54-56: the authors refer that the gene body is often more methylated than promoter regions preceding the gene. However, there are some genes in which the promoter region is more prone to accumulate CpG methylation because it would alter its expression more effectively. Please elaborate on your introduction statement, explaining the reasons why methylating gene bodies would be potentially more biologically advantageous. Lines 59-60: The authors state aberrant methylation patterns have been implicated in the carcinogenesis of mucosal warts caused by high-risk HPV infection. The references provided report data from cervical precancer specimens and not mucosal warts. Not accurate, please provide more adequate references or modify the statement. Methods section, DNA extraction: from which anatomical location were the warts biopsied? This information has implications with some of the aspects brought up in the discussion; many references presented are from oral squamous carcinoma, or high-risk HPV infection is more commonly studied using cervical specimens. All specimens were collected from males and one of the genes identified as altered is associated with male infertility. Would be very interesting to include women in this study. Was there any technical reason to only study males? Methods section, lines 77-79: “The 24 samples were then made up to approximately 500 ng of DNA in 45uL, after which they were bisulfite converted (…)” Were the samples pooled prior to the bisulfite treatment? It is not clear if the 24 samples were processed individually until sequencing. How were the samples uniquely labelled and identified/barcoded for sequencing? Figure 2: were the CpG sites outside the 0-1 coordinates flanking genomic tiling regions also quantified and included in the analysis Lines 119-129: referencing figure 3 and then figure 7 is confusing. Maybe consider reorganizing the figures. Figures 5 and 6: though these figures report to different data processing outputs, they show the same segregation between warts tissue and normal tissue; maybe opt to only present one figure. Figure 9 warrants further clarification on some collection genes present higher odds ratio, while others present a higher significance p value. What information can be retained form the two different metrics. Collections coming as significant differ from hypomethylated genes vs. hypermethylated genes. This finding warrants more discussion. Figures 10 and 12 would benefit from a better clarification. Discussion: the majority of the discussion section is a repetition from Table 2. This manuscript would benefit from possible explorations and explanations on why these differentially methylated genes would influence the development of cutaneous warts and potentially the biology of low-risk HPV types. This is the real goal and focus of this study and it requires a more in depth discussion of the biology of these viruses. Discussion, lines 215-217: the authors need to clarify this statement. In what extend knowledge of methylation profiles of benign HPV-associated warts would improve the understanding of HPV-associated carcinogenesis? Discussion section: many of the HPV-related references presented refer to oral squamous cell carcinoma. How does that relate to the anatomical location of the warts studied? Explorative comments on how previous findings from high-risk HPV types relate to this study should be included. What that tells us about the biology of papillomaviruses given that low-risk HPV types have different phenotypes and disease-associated biology.
Author Response
Dear Sir/Madam,
I would like to extend my deepest thanks to you for your constructive comments and suggestions with regard to the manuscript titled “Genome-wide tiling array analysis of HPV-induced warts reveals aberrant methylation of protein-coding and non-coding regions”. I am pleased to submit the revised version of the paper that includes a point-by-point response to the reviewer comments:
Comments by Reviewer 1
This manuscript describes a genome wide analysis in tissue samples from low-risk HPV-induced warts in comparison with surrounding normal tissue. The authors report genes that are either hypermethylated or hypomethylated in warts tissue when compared to normal tissue.
This study provides valuable insight on the epigenetics of cutaneous warts and describes significant methylation pattern differences that could shed light to the understanding of host modulation by HPV infection. The authors handled highly complexed genomic datasets analyzed by R software packages. The findings sound interesting and worth exploring its biological mechanisms, but some explanations are lacking and some figures could become supplemental due to its redundancy. Also, the manuscript would benefit from a more detailed discussion focusing on exploring explanations that would clarify the findings. This reviewer feels it would improve greatly this manuscript if the discussion section would be restructured. The study is innovative and reports good findings that would get higher visibility with explorative associations.
Lines 54-56: the authors refer that the gene body is often more methylated than promoter regions preceding the gene. However, there are some genes in which the promoter region is more prone to accumulate CpG methylation because it would alter its expression more effectively. Please elaborate on your introduction statement, explaining the reasons why methylating gene bodies would be potentially more biologically advantageous.
Elaborated on introduction statement (lines 56-62).
Lines 59-60: The authors state aberrant methylation patterns have been implicated in the carcinogenesis of mucosal warts caused by high-risk HPV infection. The references provided report data from cervical precancer specimens and not mucosal warts. Not accurate, please provide more adequate references or modify the statement.
Modified statement (line 66).
Methods section, DNA extraction: from which anatomical location were the warts biopsied? This information has implications with some of the aspects brought up in the discussion; many references presented are from oral squamous carcinoma, or high-risk HPV infection is more commonly studied using cervical specimens.
Added the anatomical locations of the wart biopsies (lines 77-78).
All specimens were collected from males and one of the genes identified as altered is associated with male infertility. Would be very interesting to include women in this study. Was there any technical reason to only study males?
As this is an exploratory study, only male subjects were included in order to reduce any gender-specific variation that might arise from sex-biased expression patterns. This limitation was included in the Discussion (lines 358-360).
Methods section, lines 77-79: “The 24 samples were then made up to approximately 500 ng of DNA in 45uL, after which they were bisulfite converted (…)” Were the samples pooled prior to the bisulfite treatment? It is not clear if the 24 samples were processed individually until sequencing. How were the samples uniquely labelled and identified/barcoded for sequencing?
The samples were not pooled prior to bisulfite treatment, and they were processed individually until sequencing. Each sample was placed in a clearly labelled microcentrifuge tube and wrapped in Parafilm. Then, each wrapped microcentrifuge tube was placed in a clearly labelled Falcon tube, the latter of which was also wrapped in Parafilm.
Figure 2: were the CpG sites outside the 0-1 coordinates flanking genomic tiling regions also quantified and included in the analysis
Yes, all CpG sites were accounted for and included in the analysis. Genomic tiling regions having more DM CpG sites and a lower combined rank score were included in the study. Details about the combined rank score can be found in the Methods section (lines 98-111).
Lines 119-129: referencing figure 3 and then figure 7 is confusing. Maybe consider reorganizing the figures.
Reorganized figures to make them more cohesive with the main text.
Figures 5 and 6: though these figures report to different data processing outputs, they show the same segregation between warts tissue and normal tissue; maybe opt to only present one figure.
Removed Figure 6.
Figure 9 warrants further clarification on some collection genes present higher odds ratio, while others present a higher significance p value. What information can be retained form the two different metrics. Collections coming as significant differ from hypomethylated genes vs. hypermethylated genes. This finding warrants more discussion.
More elaboration was added to the Results section (lines 176-192) and also to the captions of Figures 9, 10 ,11, and 12 explaining the information which can be retained from the different metrics as well as more clarification on the LOLA analysis results.
Figures 10 and 12 would benefit from a better clarification. U
Added clarification to the captions of Figures 10 and 12.
Discussion: the majority of the discussion section is a repetition from Table 2. This manuscript would benefit from possible explorations and explanations on why these differentially methylated genes would influence the development of cutaneous warts and potentially the biology of low-risk HPV types. This is the real goal and focus of this study and it requires a more in depth discussion of the biology of these viruses.
Removed Table 2 to exclude any redundancy or repetitiveness in the Discussion. However, it is difficult to pinpoint how the DM genes might influence wart development, as many of the DM genes are not well-understood. Moreover, there is a dearth of information on host DNA methylation patterns in low-risk HPV infection.
Discussion, lines 215-217: the authors need to clarify this statement. In what extend knowledge of methylation profiles of benign HPV-associated warts would improve the understanding of HPV-associated carcinogenesis?
Modified statement (lines 256-259).
Discussion section: many of the HPV-related references presented refer to oral squamous cell carcinoma. How does that relate to the anatomical location of the warts studied?
Added paragraph at the end of the discussion related to anatomical location (lines 358-360).
Explorative comments on how previous findings from high-risk HPV types relate to this study should be included. What that tells us about the biology of papillomaviruses given that low-risk HPV types have different phenotypes and disease-associated biology.
Due to differences in their expression patterns, it is difficult to directly extrapolate the results of previous findings on high-risk HPV to low-risk HPV.
Please do not hesitate to contact me if you need any additional information. I highly look forward to hearing from you.
Yours sincerely,
Dr. Laith N. AL-Eitan, MSc, PhD
Associate Professor of Human Genetics and Pharmacogenetics
Department of Biotechnology & Genetic Engineering
Faculty of Science and Arts
Jordan University of Science and Technology,
P.O.Box 3030, Irbid 22110, JORDAN
Email: lneitan@just.edu.jo
Tel.: +962-2-7201000 ext.: 23464
Reviewer 2 Report
This is a very well written paper and the standard of grammar is excellent. I could not detect any typographical errors. To my knowledge this is the first study that has looked at the level of methylation of host genes investigating the differences in methylation between HPV-induced cutaneous warts and normal skin. The methodologies are correct.
Therefore I suggest that this paper could be published at its present form.
Author Response
Dear Mr. Navarro,
I would like to extend my deepest thanks to the reviewer for taking the time to review our manuscript titled “Genome-wide tiling array analysis of HPV-induced warts reveals aberrant methylation of protein-coding and non-coding regions”.
Round 2
Reviewer 1 Report
This manuscript describes an extensive analysis of genome wide tiling in tissue samples from low-risk HPV-induced hand warts. The authors report a genome wide approach to measure methylation. This study is interesting filling a gap of knowledge about the biology of low-risk HPV infections.
This manuscript was much improved and clarified for a better reading flow, though a few more details warrant attention and should be addressed for the purpose of clarification of the study hypothesis.
Lines 50 - 53: The authors state that warts are commonly cleared by the human body in 1-2 years. Why do the authors consider epigenetic modifications may be involved as a regulator of wart formation? How these abnormal methylation levels relate to the ability of clearing skin warts?
Lines 249 - 256: this reviewer thinks the manuscript would be enhanced if an exploratory hypothesis was discussed on why these genes would be differentially methylated in a wart environment. The overall goal of this study is sound and interesting to the field, by improving our understanding of low-risk HPVs biology, though a bit more context is desirable to relate the findings to the biology of warts.
Lines 260 - 273: many of these differentially methylated genes are found deregulated in various cancers. What do the authors think would be the biological significance of the same genes differentially methylated in benign lesions?
Discussion comment on anatomical location of warts: it is very important to acknowledge that these biopsy samples were mainly from the hand skin. however, the reference included in the comment shows a different approach to explain their conclusions. The methylation profiles are independent of anatomical location in the head and neck area, but also independent from HPV status, as HPV-positive cancers show a distinct profile from HPV-negative cancers. Also, the findings described by that reference are from a cancer context; how do the authors think it relates to benign replicative warts that are commonly cleared by the human body? Would different pathways be involved to affect the growth of cancerous lesions versus a benign lesion?
Author Response
Dear Reviewer,
Much thanks to you for your constructive comments and feedback on “Genome-wide tiling array analysis of HPV-induced warts reveals aberrant methylation of protein-coding and non-coding regions”. I am pleased to submit the second revised version of the paper that includes a point-by-point response to the comments:
Comments by Reviewer 1
This manuscript describes an extensive analysis of genome wide tiling in tissue samples from low-risk HPV-induced hand warts. The authors report a genome wide approach to measure methylation. This study is interesting filling a gap of knowledge about the biology of low-risk HPV infections.
This manuscript was much improved and clarified for a better reading flow, though a few more details warrant attention and should be addressed for the purpose of clarification of the study hypothesis.
Lines 50 - 53: The authors state that warts are commonly cleared by the human body in 1-2 years. Why do the authors consider epigenetic modifications may be involved as a regulator of wart formation? How these abnormal methylation levels relate to the ability of clearing skin warts?Added the following as an answer to the aforementioned questions:
Over the course of the HPV life cycle, the viral genome undergoes dynamic changes in methylation patterns, an observation that has been tentatively attributed to the host’s innate immune response [14]. In fact, HPV methylation is the initial trigger for the transformation of squamous epithelial cells, the latter of which undergo HPV-induced epigenetic re-programming [15].
Lines 249 - 256: this reviewer thinks the manuscript would be enhanced if an exploratory hypothesis was discussed on why these genes would be differentially methylated in a wart environment. The overall goal of this study is sound and interesting to the field, by improving our understanding of low-risk HPVs biology, though a bit more context is desirable to relate the findings to the biology of warts.
Added following to relate the findings to the biology of warts:
However, EXOC4 encodes a component of the exocyst complex, the latter of which is posited to be involved in viral protein transfer between cells [36]. As part of its infection process, HPV relies heavily on membrane-bound transport vesicles to deliver the viral material to the from the extracellular matrix to the host cell’s nucleus [37–39].
KCNU1 might be important to HPV biology due to the fact that potassium channels are involved in cell proliferation and apoptosis, among other cellular processes [41].
In the context of viral infection, the deletion of RTN1 in yeast cells led to significant inhibition of viral replication [44].
Lines 260 - 273: many of these differentially methylated genes are found deregulated in various cancers. What do the authors think would be the biological significance of the same genes differentially methylated in benign lesions?
Added the following to answer the aforementioned question:
The fact that some protein-coding genes are DM in both warts and various HPV-associated cancers could potentially suggest that the extent of methylation contributes to whether the phenotype is malignant or benign.
Discussion comment on anatomical location of warts: it is very important to acknowledge that these biopsy samples were mainly from the hand skin. however, the reference included in the comment shows a different approach to explain their conclusions. The methylation profiles are independent of anatomical location in the head and neck area, but also independent from HPV status, as HPV-positive cancers show a distinct profile from HPV-negative cancers. Also, the findings described by that reference are from a cancer context; how do the authors think it relates to benign replicative warts that are commonly cleared by the human body? Would different pathways be involved to affect the growth of cancerous lesions versus a benign lesion?
Clarified that the reference was based on a cancer context, and added the following:
The hands and forehead, for example, are exposed to environmental factors that other parts of the body are not, leading to differences in gene expression between exposed and non-exposed skin [96].
Please do not hesitate to contact me if you need any additional information. I highly look forward to hearing from you.
Yours sincerely,
Dr. Laith N. AL-Eitan, MSc, PhD
Associate Professor of Human Genetics and Pharmacogenetics
Department of Biotechnology & Genetic Engineering
Faculty of Science and Arts
Jordan University of Science and Technology,
P.O.Box 3030, Irbid 22110, JORDAN
Email: lneitan@just.edu.jo
Tel.: +962-2-7201000 ext.: 23464